# Genetic Mutations Associated With TNFAIP3 (A20) Haploinsufficiency and Their Impact on Inflammatory Diseases

**DOI:** 10.3390/ijms25158275

**Published:** 2024-07-29

**Authors:** Eva Bagyinszky, Seong Soo A. An

**Affiliations:** 1Graduate School of Environment Department of Industrial and Environmental Engineering, Gachon University, Seongnam 13120, Republic of Korea; 2Department of Bionano Technology, Gachon Medical Research Institute, Gachon University, Seongnam 13120, Republic of Korea

**Keywords:** A20 haploinsufficiency, TNFAIP3 gene, haploinsufficiency, immune dysfunctions, autoimmunity, mutations, therapy

## Abstract

TNF-α-induced protein 3 (TNFAIP3), commonly referred to as A20, is an integral part of the ubiquitin-editing complex that significantly influences immune regulation, apoptosis, and the initiation of diverse immune responses. The A20 protein is characterized by an N-terminal ovarian tumor (OTU) domain and a series of seven zinc finger (ZNF) domains. Mutations in the TNFAIP3 gene are implicated in various immune-related diseases, such as Behçet’s disease, polyarticular juvenile idiopathic arthritis, autoimmune thyroiditis, autoimmune hepatitis, and rheumatoid arthritis. These mutations can lead to a spectrum of symptoms, including, but not limited to, recurrent fever, ulcers, rashes, musculoskeletal and gastrointestinal dysfunctions, cardiovascular issues, and respiratory infections. The majority of these mutations are either nonsense (STOP codon) or frameshift mutations, which are typically associated with immune dysfunctions. Nonetheless, missense mutations have also been identified as contributors to these conditions. These genetic alterations may interfere with several biological pathways, notably abnormal NF-κB signaling and dysregulated ubiquitination. Currently, there is no definitive treatment for A20 haploinsufficiency; however, therapeutic strategies can alleviate the symptoms in patients. This review delves into the mutations reported in the TNFAIP3 gene, the clinical progression in affected individuals, potential disease mechanisms, and a brief overview of the available pharmacological interventions for A20 haploinsufficiency. Mandatory genetic testing of the TNFAIP3 gene should be performed in patients diagnosed with autoinflammatory disorders to better understand the genetic underpinnings and guide treatment decisions.

## 1. Introduction: TNFAIP3 Gene, A20 Protein, and Its Functions and Importance

The TNF-α-induced protein 3 (TNFAIP3) gene, located on chromosome 6, encodes a ubiquitin-editing enzyme known as A20. This enzyme plays an important role in the regulation and induction of the immune system, including the negative regulation of nuclear factor kappa-light-chain-enhancer of activated B cells (NF-κB), ubiquitin editing, controlling immune cell differentiation, regulation of inflammatory cytokines, protection against autoimmunity, and the regulation of NLRP3 inflammasome formation. Furthermore, A20 is also associated with the regulation of cell death (apoptosis/necrosis), and as a tumor suppressor, it regulates cell proliferation (Figure 1) [1,2,3,4,5,6].

A20 protein is highly conserved, with two main domains: an N-terminal ovarian tumor (OTU) domain and seven zinc finger (ZNF) domains in the C-terminal area (Figure 2 and Figure 3) [4]. The OTU domain of A20 is a cysteine protease domain involved in deubiquitination-related pathways, with the catalytic residues Asp70, Cys103, and His256, where the highly conserved Cys103 is crucial for ubiquitin cleavage. By recognizing and cleaving specific ubiquitin chains, the OTU domain can regulate inflammatory signaling pathways. The OTU can interact with E2 enzymes (Ubc13 and UbcH5c) and regulate their functions, recognizing both Lys-48- and Lys-63-linked ubiquitin chains. The Lys-48 ubiquitin chain recognized by OTU is associated with proteasomal degradation, while cleaving Lys-63-linked ubiquitin chains could impact immune signaling pathways [4,5]. 

The A20 protein contains seven ZNF domains responsible for protein stabilization and protein–protein interactions. The ZNF4 domain plays an important role in immune regulation by interacting with TNF receptor-associated factor 6 (TRAF6), leading to the inhibition of pro-inflammatory pathways such as the NF-κB pathway. ZNF4 was also suggested to have ubiquitin-binding activity by interacting with E2 enzymes and has E3 ubiquitin ligase activity. The ZNF4 domain can interact with the seventh ZNF domain and can bind methionine 1 (M1)-linked ubiquitin, which restricts the IκB-gamma kinase (IKK-gamma) pathways. ZNF domains have several phosphorylation sites, including Ser381, Ser480, Ser565, and Thr625, which are critical for Lys48- and Lys63-related ubiquitin activities [4,5,6,8].

Heterozygous mutations in TNFAIP3 are associated with loss-of-function mechanisms. Mutant A20 protein results in inhibition of NF-κB signaling, leading to elevated expression of pro-inflammatory molecules (such as IL-1β, IP-10, sTNFR1, IL-6, or TNF-α), which is associated with autoinflammatory diseases [9,10]. Several mutations have been described in the TNFAIP3 gene. The majority of disease-related variants are either frameshift or STOP codon variants, but a few missense mutations have also been discovered (https://infevers.umai-montpellier.fr/web/search.php?n=26, accessed on 20 March 2024).

Disease phenotypes can be variable and include Behcet’s disease (BD), polyarticular juvenile idiopathic arthritis, autoimmune thyroiditis, autoimmune hepatitis, rheumatoid arthritis. Disease symptoms can also be variable, with the most common forms being recurrent fever or rashes [6,10,11]. However, other symptoms, including musculoskeletal dysfunctions, gastrointestinal dysfunctions, cardiovascular symptoms, or respiratory tract infections, have also been identified. The age of onset varies widely, as both child- and adult-onset forms of the disease have been reported [11,12]. Rarely, the disease can involve the central nervous system or blood vessels [6,12].

Haploinsufficiency is a loss-of-function mechanism associated with heterozygous mutations, especially STOP codon or frameshift variants. It usually occurs in genes that need to be expressed in large amounts for appropriate functioning. Haploinsufficiency can also be caused by mutations in genes with regulatory functions. In the case of haploinsufficiency, one normal allele cannot produce enough protein for proper cell functioning. At least 300 diseases have been described as being caused by haploinsufficiency, including epilepsy, Whipple’s disease, Autoimmune Lymphoproliferative Syndrome, and autism. The majority of these diseases have autosomal dominant inheritance, but heterozygous forms of autosomal recessively inherited diseases have also been suggested to be associated with milder forms of haploinsufficiency [13,14,15,16,17,18,19].

Different methods are available to detect reduced mRNA levels in possible cases of haploinsufficiency. To verify whether a mutation could reduce mRNA levels in patients, real-time quantitative polymerase chain reaction (qPCR) is an easy method to detect mRNA levels in patients’ blood or other tissues (including brain, tumor, etc.) [18,20]. RNA sequencing is a high-throughput analysis used to screen the gene expression patterns of cells. Besides quantifying gene expression, it can also be useful in detecting other alterations, including alternative splicing, alterations of noncoding RNA, and microRNA patterns [21,22]. Reduced mRNA levels in the case of haploinsufficiency result in lower protein levels. Western blot and flow cytometry have also been verified as useful methods to quantify proteins from human tissues and predict whether a mutation could be associated with low gene dosage [23,24].

In this review, we summarize the immune disease-related mutations described in the coding region of the TNFAIP3 gene, including the phenotypes and disease progression. Furthermore, we discuss the possible disease mechanisms associated with TNFAIP3 mutations and introduce some examples of treatment strategies.

## 2. Mutations in TNFAIP3 Gene and Disease Phenotypes

### 2.1. Large Deletions

Several mutations have been described in the TNFAIP3 gene, leading to A20 haploinsufficiency-related diseases, including frameshift, splice site, missense, and STOP codon mutations (Table 1, Table 2 and Table 3). Furthermore, smaller or larger deletions in chromosome 6 were identified, which resulted in full or partial deletion of TNFAIP3 gene (Table 1).

To date, 12 cases of large deletion relation-related A20 haploinsufficiency have been reported. The majority of cases were associated with de novo deletions. Most patients with large deletions were associated with a young onset, with more than 80% of patients presenting disease phenotypes before their first year. The most common initial symptom was fever, but stomatitis, autoimmune hemolytic anemia, and ulcers were also described. Rare symptoms could include intellectual disability or dysmorphic features [25,26,27,28,29,30].

**Table 1 ijms-25-08275-t001:** Large deletions in TNFAIP3 associated with A20 haploinsufficiency and other immune diseases.

Mutation	Age of Onset	Domain	Symptoms	Family History	Final Diagnosis	Reference
Large deletion	Infancy	Entire gene	Episodes of fever, restricted growth, enlarged liver/spleen	Negative	Inflammatory and other syndromic manifestations	[25]
Large deletion in 6q	Young onset	Entire gene	Recurring high fever, chronic systemic lymphadenopathy	NA	Autoimmune lymphoproliferative syndrome	[26]
Large deletion in chr6	2 months	Entire gene	Periodic fever, abdominal pain, diarrhea, bloody stools, folliculitis	Positive	A20 haploinsufficiency	[27]
Exon 2–3 deletion	6 years	Partial deletion	Colic with fever, headache, vomiting, and oral/perianal ulcers	Negative	BD-like A20 haploinsufficiency	[29]
Exon 7–8 deletion	10 months	Partial deletion	Fever, elevated leukocytes, lymphadenopathy, pericardial effusion, persistent hepatosplenomegaly, abnormal kidney and liver function	Negative	Lupus nephritis	[30]
TNFAIP3 microdeletion	Infancy	Entire gene	Intermittent high fever, diarrhea, developmental delay, oral aphthous ulcers, arthritis, and intestinal punched-out ulcers	Negative	Inflammatory bowel disease	[28]

**Table 2 ijms-25-08275-t002:** Examples of splice site, frameshift, or nonsense variants in TNFAIP3 associated with A20 haploinsufficiency and other immune diseases.

Mutation	Age of Onset	Domain	Symptoms	Family History	Final Diagnosis	Reference(s)
c.1906 + 1G > C	10 years	Splice site	Lower back pain, thoracic kyphosis and recurrent oral ulcers, dwarfism	Negative	A20 haploinsufficiency	[31]
c.1906+2T>G	15 years	Splice site	Oral and genital ulcers, folliculitis, proteinuria	Positive	A20 haploinsufficiency	[32]
Trp85Glyfs*11	11 years	OTU	Fever, polyarthritis	Positive	Behçet’s disease	[33]
Cys86Trp fs*8	6 years	OTU	Acneiform lesions, retinal vasculitis, nonerosive and asymmetric arthritis, pulmonary issues, anemia, hepatitis	Positive	A20 haploinsufficiency	[34]
Arg87Ter	3 weeks–43 years	OTU	Ulcers; autoimmune features such as type I diabetes, Hashimoto thyroiditis, and pernicious anemia; abdominal pain	Positive	A20 haploinsufficiency	[35]
	7 years	OTU	Fever, chronic joint synovitis	Positive	Polyarticular juvenile idiopathic arthritis	[30]
Lys91Ter	Young onset	OTU	Autoimmune thyroid disease, juvenile idiopathic arthritis, psoriasis, liver disease, and immunodeficiency	Positive	A20 haploinsufficiency	[36,37]
Asn98Thr fs25*	2 years	OTU	Oral and genital ulcers	Positive	Behcet’s disease-like syndrome	[38]
Leu147Glnfs∗7	2 years	OTU	Infectious enteritis and stomatitis	Negative	Behcet-like autoinflammatory syndrome	[39]
Glu154Ter	1 year	OTU	Normocytic anemia, hematochezia	NA	Inflammatory colitis	[40]
Trp164Ter	3 months	OTU	Recurrent fever, erythematous rashes, oral ulcers, SLE-like disease	Positive	A20 haploinsufficiency	[41]
Arg183Ter	2 years	OTU	Fever, perianal abscesses	NA	A20 haploinsufficiency	[31]
	5 years	OTU	Fever, mucosal ulcers, pustules and pustular pooling (co-existed with IL36RN mutation)	Positive	Behçet-like autoinflammatory syndrome	[42]
Gln187Ter	7 years	OTU	Hepatomegaly liver fibrosis, anemia, fever, rash	Positive	Lupus erythematosus, lymphoproliferation	[43]
	7 years	OTU	Abdominal swelling, intermittent fever	Positive	Hepatic fibrosis, pericardial effusion, and hypothyroidism	[44]
Cys200Ala fs*16	3 years	OTU	Oral ulcer and epigastralgia, fever	Positive	Autoinflammatory disease	[45]
Arg204fs	3 years	OTU	NA	NA	Refractory diarrhea	[46]
Asp212Gly fs*38 (c.634+2T>C)	29 years	OTU	Alopecia, shoulder pain, proteinuria, mild thrombocytopenia, positive autoantibodies	Positive	Lupus nephritis	[30]
Leu218Trp fs*10	3 years	OTU	Toothache, fever, skin rash, lymph node enlargement	Positive	A20 haploinsufficiency	[47]
Phe224Ser fs*4	28–61	OTU	Oral/genital ulcers and polyarthritis, lupus erythematosus, with CNS vasculitis	Positive	Early-onset autoinflammatory syndrome	[48]
Pro226Leu fs*2	Infancy	OTU	Periodic fever, abdominal pain, and vomiting	Negative	A20 haploinsufficiency	[49]
Leu227Ter	Early onset	OTU	Systemic inflammation, arthralgia/arthritis, oral/genital ulcers, and ocular inflammation	Positive	Early-onset autoinflammatory syndrome	[48]
Pro268Leufs*19	NA	OTU	Autoinflammatory disease, co-existed with pathogenic MEFV variants	Familial	A20 and familial Mediterranean fever	[49]
Arg271Ter	10	OTU	Oral ulcers, perianal ulcers with abscesses	Positive	A20 haploinsufficiency	[32]
	5	OTU	NA	NA	Behcet’s disease-like syndrome	[38]
Leu303fs*26	4	OTU	Intermittent abdominal pain, diarrhea, bleeding and inflammation in ileum and colon, short statue	Positive	A20 haploinsufficiency	[34]
	5 years	OTU	Chronic abdominal pain, aphthous stomatitis, loss of appetite, vomiting, malnutrition, perianal lesions, knee joint swelling, bloody stool	Negative	Pediatric Crohn’s disease	[50]
Tyr306Ter	5 years	OTU	Mild undifferentiated colitis, genital and oral ulcers	Positive	A20 haploinsufficiency	[48]
Val309dup	3 months	OTU	Swelling and pain in multiple joints, fever, diarrhea	NA	A20 haploinsufficiency	[31]
Gln370Argfs*16.	Infancy	Between OTU and ZNF1	Oral ulcer, diarrhea, hematochezia, rash, arthritis	NA	Inflammatory bowel disease	[51]
Ser408Ter	1.5 years	ZNF1	Recurrent fever	Negative	A20 haploinsufficiency	[52]
Gln415fs	7 months	Between ZNF1 and ZNF2	Fever, liver dysfunction, cervical lymph node swelling, and skin rash	Negative	Autoimmune lymphoproliferative syndrome, Kawasaki disease	[53]
Ala434Ter (c.1300_1301 delinsTA)	3 years	Between ZNF1 and ZNF2	Intermittent fever, frequent oral ulcers, hyperemic rash, hemolytic anemia, proteinuria, hematuria	Positive	Lupus nephritis	[30]
Cys478Ter	6–12 years	ZNF2	Nausea, fever, lymphadenopathy, ulcers, erythema nodosum	Positive	BD-like syndrome	[54]
Val489Alafs*7	10 years	ZNF2	Diabetes, cytopenia, hepatitis, enteropathy, and interstitial lung disease	Positive	Early-onset autoimmune disease	[55]
Gln490Ter (c.1467_1468 delinsAT)	19 years	ZNF2	Diabetes, ulcers, liver dysfunction	Negative	Diabetes and Behcet’s-like disease	[56]
Lys564Ter	1.5 and 21 years	ZNF3	Fever, intermittent abdominal pain, esophageal and ileocecal ulcers, BD-like syndrome, short statue	Positive	A20 haploinsufficiency	[34]
His577Alafs*95	9–41 years	Between ZNF3 and ZNF4	Thyroiditis, type I diabetes, hemolytic anemia, and chronic polyarthritis	Positive	Poly-autoimmunity	[57]
Gln593Ter	1 year	ZNF3–ZNF4	Atopic dermatitis, cervical lymphadenopathy, fevers, headache, and neck pain	NA	A20 haploinsufficiency and lupus	[58]
Thr602fs*95	32	ZNF4	Cough, headache, slow response, short-term memory loss, bilateral ptosis, myalgia of limbs, ulcers	Negative	Systemic lupus erythematosus	[59]
Thr604Argfs*93	28–64	ZNF4	Gastrointestinal ulcerations	Positive	A20 haploinsufficiency	[48]
Leu626Valfs*45	8 years	ZNF4	Liver dysfunction	NA	Autoinflammatory liver disease	[60]
His636fs*1	16 years	ZNF4	High fever and severe abdominal pain, anorexia, inflammatory polyarthritis affecting the large joints		Adult-onset Still’s disease	[61]
Glu730Ser fs*83	1 months and 10 years	ZNF5	Fever, cerebral infarction, diarrhea, lung dysfunctions	Positive	A20 haploinsufficiency and lupus	[34]
Gln737Ser fs*79	6 months–28 years	ZNF6	Abdominal pain, vomiting, and bloody stools	Positive	Hashimoto’s thyroiditis	[62]
Lys759Serfs*56	5 months	ZNF7	Recurrent fever, hematochezia, and steatorrhea	Positive	Behçet’s disease	[63]

**Table 3 ijms-25-08275-t003:** Missense mutations in TNFAIP3 associated with A20 haploinsufficiency and other immune diseases.

Mutation	Age of Onset	Domain	Symptoms	Family History	Final Diagnosis	Reference
Asn102Ser	29	OTU	Ulceration of intestinal anastomosis, oral ulcers, and vasculitis in extremities		Intestinal Behcet’s disease	[24]
Thr129Met	3 months	OTU	Fever, irritability, pallor, and hepatosplenomegaly	Positive	Hemophagocytic lymphohistiocytosis	[64]
Glu192Lys	11 months	OTU	Cervical lymphadenitis, repeated tonsillitis, cervical lymphadenitis, and bronchitis	Positive	Atypical inflammatory disease	[65]
Leu236Pro	13 years	OTU	Abdominal pain, diarrhea, colonic and oral ulcers, musculoskeletal symptoms	Positive	A20 haploinsufficiency	[66]
Cys243Tyr	17 years	OTU	Oral and genital ulcers, erythema nodosum-like lesions	Positive	Behçet’s disease	[67]
Pro247Leu	4 years	OTU	Fever, vomiting, abdominal and joint pain, swelling, limited movement	De novo	A20 haploinsufficiency	[68]
Leu275Pro	3 months	OTU	Fever and hepatosplenomegaly	NA	Hemophagocytic lymphohistiocytosis	[65]
Ile310Thr	20 years	OTU	Severe abdominal pain, recurrent fever	Negative	Atypical inflammatory disease	[65]
Thr474Ala	12 years	ZNF2	Mild epistaxis, multiple episodes of thrombocytopenia	Negative	Chronic immune thrombocytopenia and vitiligo	[69]
Met476Ile	13 years	ZNF2	Intermittent fever, diffuse lymphadenopathy, arthritis, and recurrent multiple gastrointestinal ulcers	NA	Behçet-like disease, persistent EBV viremia	[70]
Ala547Thr	18 years	ZNF3	Recurrent fever, joint pain, osteomyelitis, oral and genital ulcers	Positive	A20 haploinsufficiency	[32]
Thr602Ser	NA	ZNF4	Oral ulcers, skin rashes, eye conjunctivitis, perianal ulcers, and bowel inflammation	Positive	Behçet-like disease	[71]
Thr647Pro	8 years	ZNFf4	Focal seizures and hemiparesis, uveitis, cognitive decline, mouth ulcers	Positive	A20 haploinsufficiency	[72]
Gln709Arg	24 years	ZNF5	Fever, facial erythema, proteinuria, lupus-like symptoms	Positive	Atypical inflammatory disease	[65]

A 119 kb (6q23.3: 138125829_138244816; GRCh37) microdeletion was also observed in chromosome 6q, which included the TNFAIP3 gene. This microdeletion was reported in a case of infantile onset inflammatory bowel disease (IBD), severe perianal lesions, hepatosplenomegaly, and high C-reactive protein levels. The patient was 8 months old and had no family history of the disease. Her growth was impaired, but no additional developmental delay was observed. The patient had BD-like symptoms, such as aphthous ulcers, intestinal punched-out ulcers, and arthritis. However, the patient did not fulfill the criteria for a BD diagnosis, and she was diagnosed with IBD with perianal lesions. As treatment, she received colchicine and 5-aminosalicylic acid. Surgery was also performed, and to improve the fistulae, setons were implanted [25,27].

Partial deletions of TNFAIP3 have also been reported. A patient with a deletion of exons 2–3 was a 9-year-old child who presented several symptoms, including abdominal colic, fever, headache, vomiting, and ulcers in the abdominal and perianal regions. Ulcers were also observed in the cecum and ascending colon. The initial diagnosis was intestinal BD, but the diagnosis was revised to A20 haploinsufficiency after genetic testing. Histopathology showed mild neutrophil infiltration, lymphoid follicles, and granulation. Treatment with infliximab initially improved the symptoms; however, after 6 months, the recurrent fever, ulcers, and abdominal pain returned. The treatment was changed to adalimumab, which seemed to be effective, and the patient is currently in remission. The deletion of exons 2–3 of TNFAIP3 results in the partial loss of the OTU domain, leading to reduced expression of the A20 protein [29].

Another partial deletion in TNFAIP3 included exons 7–8 of the gene, which was associated with a de novo case of A20 haploinsufficiency in a child who experienced liver dysfunctions, persistent hepatosplenomegaly, fever with elevated leukocytes, lymphadenopathy, and massive pericardial effusion before her first year of age. She was initially diagnosed with interstitial lung disease. Later, she developed vasculitic rashes, leukopenia, and hemolytic anemia. The patient was treated with methylprednisolone pulse therapy, methylprednisolone with cyclosporine, and belimumab. The therapy was effective, and the patient is currently symptom-free. This mutation results in the loss of the C-terminal OTU domain and the first five ZNF domains [30].

### 2.2. Frameshift, Splice Site, and Nonsense Mutations

Several heterozygous frameshift and nonsense mutations were found in the TNFAIP3 gene, leading to premature STOP codons (Table 2, Figure 2).

A splice site mutation, c.1906+1G>C, is located between the ZNF3 and ZNF4 domains. This mutation was observed in an 11-year-old child with recurrent lower back pain and recurrent ulcers. The patient had developmental delay, scoliosis, thoracic kyphosis, positive spinous process tenderness, and arthritis. There were no joint issues or disturbances of the digestive system. MRI analysis showed inflammatory changes, and she was diagnosed with juvenile idiopathic arthritis. Oral sulfasalazine temporarily improved her back pain. Her diagnosis was revised to systemic lupus erythematosus, and she was treated with methylprednisolone, hydroxychloroquine, and sulfasalazine. However, the back pain became stronger after treatment was discontinued. Small pyramidal changes were observed in her at that time. She was treated with sulfasalazine and infliximab, which improved her joint symptoms. Furthermore, infliximab treatment abolished her back pain. Sequence analysis confirmed the TNFAIP3 mutation, and she was diagnosed with A20 haploinsufficiency with dwarfism [31].

Another splice site mutation in the same region, c.1906+2T>G, was observed in a Chinese family. The proband patient experienced folliculitis and oral and genital ulcers since her teenage years, without fever, polyarthritis, or uveitis. In her 30s, she developed proteinuria. The initial diagnosis was nephrotic syndrome, and the proteinuria was successfully treated with tacrolimus. Furthermore, thalidomide treatment improved her ulcers and folliculitis. The proband’s 13-year-old daughter had inflammation-related symptoms, such as ulcers, abdominal pain, and recurrent fever, but no polyarthritis or folliculitis. She was successfully treated with adalimumab and is currently asymptomatic. Other relatives had recurrent oral ulcers without fever, genital ulcers, abdominal pain, or folliculitis [32].

The Trp85Glyfs*11 mutation was observed in a Japanese family who developed BD at a young age. The proband was initially diagnosed with idiopathic arthritis before her first year of age. At the age of 4, she presented with slight arthralgia, recurrent aphthous stomatitis, aphthous ulcers, and hemorrhoids. At 16 years of age, she developed muscle pain and ulcers in the genital and colon systems. Methotrexate therapy and treatment with non-steroidal anti-inflammatory drugs (NSAIDs) were unsuccessful. Prednisolone treatment was combined with tocilizumab, which significantly improved the patient’s condition, but mild symptoms still remained. Replacement of tocilizumab with adalimumab resulted in further improvement in her condition. Other family members (the proband’s mother and grandfather) were also diagnosed with BD, but her mother developed a relatively mild disease phenotype [33].

The Cys86Trpfs*8 mutation was observed in a patient with A20 haploinsufficiency at the age of 6 years. The patient did not have fever, gastrointestinal dysfunctions, or genital ulcers, but she had several lesions, including acneiform lesions, pseudofolliculitis, and erythema nodosum. Additionally, the patient had ocular dysfunctions, including retinal vasculitis and macular degeneration. Furthermore, the patient experienced additional issues, such as pulmonary disorders (acute pulmonary hemorrhage, ventilatory dysfunction, left lung nodule), anemia, lymphopenia, thrombocytosis, hepatitis, and short stature. Multiple drugs were used in her treatment, such as prednisolone, cyclophosphamide, methotrexate, and azathioprine, which resulted in improvement in her condition. She probably had a positive family history of the disease [34].

Arg87Ter was reported in a large family. Besides the typical A20 haploinsufficiency symptoms (aphthous ulcers, autoimmune symptoms, abdominal pain), three out of the eight patients had liver involvement, which is rare among patients with A20 haploinsufficiency. Liver biopsy of the patients with liver involvement showed hepatic fibrosis, hepatocyte injury, and/or CD4+/CD8+ T cell infiltration. Treatment was still under debate at the time of publication, but one family member’s liver functions were normalized after being treated with steroids and azathioprine, followed by adalimumab [35].

Lys91Ter was reported in a family where the affected relatives presented autoimmune thyroiditis in the early disease stages at young ages. The proband patient had different symptoms, such as psoriasis, articular symptoms, atrophic gastritis, severe autoinflammatory lung reaction, anemia, and genital papillomatosis. Other affected relatives presented liver failure and polyarticular juvenile idiopathic arthritis. The lung dysfunctions and anemia were successfully treated with anakinra (IL-1 antagonist) and mycophenolate, respectively [20,36,37].

Asn98Thr fs25* was reported in a 2-year-old Caucasian patient with BD, but no further details were mentioned on the patient’s clinical symptoms [38].

The proband patient with Leu147Glnfs*7 was a 2-year-old child who had recurrent fever, diarrhea, vomiting, and stomatitis at the age of 3 months with intervals of 1–5 months. Several disease markers, such as procalcitonin (PCT), C-reactive protein (CRP), and white blood cell (WBC) count, were elevated in her blood. She was allergic to multiple foods, such as milk, eggs, and seafood. An ultrasound of her gastrointestinal system showed thickening in her colon area, which was probably due to inflammation. Lung CT revealed granulomas. The inflammation also affected her knee joints. She was diagnosed with Behcet-like autoinflammatory syndrome. The treatment included prednisone with iron supplements, but later it was changed to oral prednisone and injections of recombinant human type II tumor necrosis factor receptor–antibody Fc fusion protein. Except for the gastrointestinal issues, her condition improved. Later, she received anti-inflammatory therapy. The treatment was successful since her colon, lungs, and knee joints returned to normal [20,39].

The patient with Glu154Ter was a 1-year-old child who was diagnosed with early-onset inflammatory colitis. The initial symptoms were severe normocytic anemia and hematochezia; the later symptoms were anemia due to gastrointestinal bleeding. Later, cutaneous inflammation and Behçet-like phenotypes appeared, and the patient’s family history was positive. She initially received oral steroids with oral iron supplements. During her steroid withdrawal period, acute panniculitis appeared in her limbs. The treatment was revised to anti-inflammatory therapies. Two days later, her skin condition worsened, and edema appeared in her limbs. She received intravenous steroid therapy, which was not effective. After a genetic test, steroid therapies were stopped, and the patient received infliximab and methotrexate, which seemed to improve her symptoms. However, new skin lesions appeared with arthralgia, and the therapy was switched to adalimumab with methotrexate. The patient is currently asymptomatic [40].

The Trp164Ter mutation was found in a case of an infant with neonatal lupus erythematosus-like A20 haploinsufficiency. She presented with recurrent fever, erythematous rashes, and ulcers in her oral system. Inflammatory markers and liver enzymes were elevated in her. These disease phenotypes still remained at the age of 6 months. The family history seemed to be positive since her father also presented Behcet-like symptoms starting from his teenage years [41].

Arg183Ter appeared in a 4-year-old child who presented with intermittent fever and perianal abscesses since his first year of age. Later, he also developed recurrent fever (lasting from 3 days to 2 months) and oral ulcers. He had abdominal pain but no other gastrointestinal dysfunctions. Treatments for tropical diseases, antibiotics, and steroids were ineffective. Later, the child was diagnosed with a bacterial infection, and antibiotic and steroid therapy were useful. After a genetic test, the steroid therapy was discontinued, and thalidomide immunosuppressive therapy with hormones was given to him, which improved his condition, resulting in the discontinuation of hormones. Hi s family history may be positive since his mother also presented with oral and genital ulcers [31]. The second patient with the TNFAIP3 Arg183Ter mutation was a 5-year-old Chinese boy and he also carried another pathogenic mutation in the IL36RN gene (c.115+6T>C, Arg10Argfs*1). The child had a geographic tongue (GT) from childbirth and had recurrent fever, mucosal ulcers, and erythema. The patient was a unique case of generalized pustular psoriasis (GPP) with Behcet-like autoinflammatory syndrome. Several treatment strategies were used, such as antibiotics or intravenous immunoglobulin, which improved the fever [42].

Gln187Ter was found in a Chinese family. The first patient developed abdominal swelling at the age of 7, and her ultrasound revealed hepatomegaly, ascites, and pericardial effusion. Furthermore, abnormal liver and thyroid functions were also observed in her. At the age of 11, her liver issues continued, and she also developed intermittent fever. Her biomarker test revealed an elevated erythrocyte sedimentation rate and C-reactive protein level and reduced complement levels. An anti-TNF drug (etanercept), combined with other drugs (such as hydroxychloroquine, mycophenolate mofetil, and prednisolone), seemed to be effective for her. Her family history seemed to be positive since her brother developed juvenile idiopathic arthritis (JIA) and had Crohn’s disease and an inflammatory bowel system-like pathology. Also, their father presented symptoms of recurrent arthralgia and anal fistulae from his childhood [44]. The second case of Gln187Ter was observed in a patient with a probable positive family history of the disease. She was diagnosed with systemic lupus erythematosus with fever, rashes, hepatomegaly, and thrombocytopenia. Her biopsy showed liver fibrosis. Initially, the patient was given hydroxychloroquine, glucocorticoid, and immunosuppressive drugs (including cyclosporine and mycophenolate mofetil). Later, etanercept was added to her treatment. The therapy seemed to be successful since her symptoms improved significantly [43].

Cys200Ala fs*16 was reported in a familial case of juvenile onset autoinflammatory disease. The proband patient developed periodic fever and oral aphtha in his teenage years. Later, epigastralgia and oral ulcers also appeared, leading to poor food intake. Biomarker analysis revealed elevated CRP levels. Prednisolone was used as the treatment, which resulted in improvement in the patient’s condition. His mother and sister also presented Behçet’s disease-like symptoms [45].

The Arg204fs mutation was found in a patient with very-early-onset refractory diarrhea. The proband patient was a 3-year-old child, but no further details were mentioned on the disease progression or clinical symptoms [46].

Asp212Gly fs*38 is caused by a splice site mutation (c.634+2T>C). The patient developed familial systematic lupus erythematosus in his late 20s, with a positive family history of the disease. His symptoms were alopecia, pain in the shoulder, proteinuria, and mild thrombocytopenia. The patient had reduced serum complement levels and high IgE levels. He presented allergies to several foods, such as milk, seafood, eggs, mold, and pollen. His treatment included prednisone combined with other drugs, such as Tripterygium wilfordii compounds or tacrolimus. Significant improvement was seen in his condition after treatment, as the proteinuria, negative autoantibodies, and renal dysfunctions disappeared, and complement molecule levels returned to normal [30].

Leu218Trp fs*10 was observed in a 3-year-old child who initially experienced recurrent fever, toothache, and skin rashes. He also had generalized lymphadenopathy. His inflammatory markers and C-reactive protein levels were high. Prednisone improved his fever, but the ulcers remained. However, methotrexate and thalidomide, with reduced levels of prednisone, were successfully used against the ulcers. His father was diagnosed with non-functional pituitary macroadenoma and hypopituitarism, while his mother had Mediterranean anemia [47].

Phe224Ser fs*4 was found in a European–American family, where affected relatives developed the disease between 5 and 10 years of age. All of them had ulcers in the oral or genital areas, and skin rashes also appeared (such as folliculitis or erythematous papules). One of the patients also developed retinal vasculitis, CNS vasculitis, migraine, and colon ulcers. Patients with this mutation had an abnormal response to NF-α and IL-1R blockade [48,60].

A Pro226Leu fs*2 patient was discovered in a de novo case, where the proband patient presented with periodic fever, abdominal pain, and vomiting. Treatment with anti-TNF alpha was initially unsuccessful, but combining it with prednisolone resulted in improved symptoms [49].

Pro268Leufs*19 was observed in a Turkish family, where the affected family members presented with Mediterranean fever (FMF) and Behcet-like phenotypes. The proband was diagnosed with BD but had atypical phenotypes too, such as vesicular eruptions and Hashimoto thyroiditis. The daughters of the proband had ulcers in the oral and genital areas, but they did not fulfill the diagnostic criteria for BD. They did not present with Hashimoto thyroiditis, even though their markers were positive for the disease. The patients also carried two FMF-related gene mutations: MEFV p.Met680Ile and p.Arg761His. The proband and several family members showed a positive reaction to colchicine treatment [73].

The proband for Arg271Ter was a Chinese patient who had recurrent oral and perianal ulcers since the age of 10. In his 30s, he developed abdominal pain, intermittent fever, and diarrhea. An endoscopy analysis revealed inflammatory bowel disease, but he was also diagnosed with Hashimoto’s disease. His family history was positive [32]. Arg271Ter was also found in a Caucasian case of BD, who developed the disease at the age of 5. The patient had oral and genital aphthosis, but skin involvement was also observed. Treatment with hydroxychloroquine, azathioprine, and mycophenolate mofetil was unsuccessful [38].

Leu303fs*26 was associated with a Chinese case of A20 haploinsufficiency, where the affected patients developed the disease at the age of 5 or 6 years. They did not have fever or genital ulcers, but they developed oral ulcers. Both of them had a short stature and developed non-erosive and asymmetric arthritis. Gastrointestinal involvement may be possible since one of them had intermittent abdominal pain with diarrhea and inflammation of the colon. The other patient had retinal vasculitis and acneiform lesions. The treatments included mesalazine, prednisone, methotrexate, and TNF-alpha inhibitor [34].

The second case of Leu303fs*26 was described in a de novo case of Crohn’s disease. The patient experienced knee joint swelling, aphthous stomatitis, loss of appetite, vomiting, malnutrition, perianal lesions, bloody stool, and chronic abdominal pain. Treatment started with nutritional support, followed by hormone therapy, 5-aminosalicylic acid (mesalazine), and methotrexate. The treatment relieved the abdominal pain and malnutrition. The autoimmune dysfunctions were also reduced, but joint symptoms were still present at the time of publication [50].

Duplication of Valine309 was observed in a 3-year-old child who presented with arthritis and inflammatory bowel disease. She had limited mobility due to joint swelling and pain, especially in her right ankle. Digestive issues (such as diarrhea) and intermittent fever were also present. Colonoscopy revealed colonic ulcers. The patient underwent synovectomy and joint irrigation surgery, which were ineffective. However, her symptoms improved after treatment with thalidomide and adalimumab [31].

Gln370Argfs*16 was observed in a child who was suspected of having inflammatory bowel disease at the age of 60 days. She had oral ulcers, diarrhea, hematochezia, rashes, rotavirus enteritis, and arthritis in major joints. Several drugs were used, such as mesalazine, intravenous immunoglobulin, and infliximab, but no surgery was performed. The patient’s condition was stable at the time of publication [51].

Ser408Ter was found in a child without any family history of disease. The patient experienced recurrent fever starting from 1.5 years of age. No developmental, eye, skin, or digestive issues were observed. CRP and serum amyloid A levels were elevated during fever periods. Colchicine resulted in an improvement in the fever [52].

Gln415fs was observed in a 7-month-old Japanese child who developed a fever. His C-reactive protein levels were high. Later, additional symptoms appeared, including liver dysfunctions, cervical lymph node swelling, and skin rash. The patient was diagnosed with Kawasaki disease. Intravenous immunoglobulin infusion therapy was not successful. In his blood, elevated pro-inflammatory markers, such as IL-10 and IL-18, were detected. Liver biopsy showed hepatitis, spotty focal necrosis, and giant multinucleated hepatocytes, and skin biopsy specimens revealed CD4+ and CD8+ T cell infiltration. The final diagnosis of the patient was autoimmune lymphoproliferative syndrome, and he was treated with prednisolone and cyclosporine. Except for the recurrent liver issues, the treatment was successful. However, the combined therapy with mycophenolate mofetil and prednisolone eliminated the liver issues [53].

Ala434Ter (c.1300_1301 delinsTA) was found in a Chinese child with autoimmune lupus nephritis. Her family history seemed to be positive. Her symptoms appeared at the age of 3, and her symptoms were recurrent fever, oral ulcers, bilateral knee effusion, autoimmune hemolytic anemia, and renal dysfunctions. Further features were swollen lymph nodes and hepatosplenomegaly. Complement markers (C3 and C4) were reduced, and the patient was positive for autoantibodies. She received different therapies, including methylprednisolone pulse therapy, prednisone, hydroxychloroquine, and tofacitinib, which resulted in complete remission [30].

Cys478Ter was observed in two siblings. The first patient was diagnosed with BD at the age of 7 but had fever and lymphadenopathy at the age of 6. He also had oral and perianal ulcers, pharyngalgia, and nausea. His sister developed oral ulcers at the age of 12 and genital ulcers three years later. At the age of 19, she developed a fever and was diagnosed with BD. At the age of 25, she had a fever and erythema nodosum, and treatment was unsuccessful. She also had ulcers in her colon and stomach, and in her 30s, she developed hepatosplenomegaly, lymphadenopathy, and pernio-like rashes. Prednisolone and colchicine were used in their treatment. Other family members, such as their brother and father, also developed similar disease phenotypes [54].

Val489Alafs*7 was observed in a child with early-onset autoimmune disease, who presented different autoimmune-related symptoms at the age of 10, including diabetes, cytopenia, hepatitis, enteropathy, and interstitial lung disease. None of his relatives presented with similar disease phenotypes. Hematopoietic stem cell transplantation resulted in significant improvement in his condition [55].

Gln490Ter (c.1467_1468delinsAT) was found in a patient who had liver dysfunctions and minor mouth ulcers since early childhood, and he was diagnosed with Type-1 diabetes at the age of 19. The patient did not experience ketoacidosis, and insulin therapy was successful. He also received ursodeoxycholic acid, which improved his liver functions [56].

Lys564Ter was reported in a family where the affected members developed the disease between 1 month and 10 years of age. They experienced intermittent or periodic fevers and oral ulcers but not genital ulcers. One of them had cerebral thrombosis and infarction, while another patient had intestinal issues, such as diarrhea and inflammation in the colon and ileum. Pulmonary and hematological abnormalities may have also been present. Prednisone, thalidomide, and azathioprine were used in treatment [34].

His577Ala fs*95 was reported in an Italian family, where the mother and three children developed poly-autoimmunity. The affected family members developed their first symptoms before 10 years of age. All affected members experienced recurrent ulcers, thyroiditis, type I diabetes, hemolytic anemia, and chronic polyarthritis. Recurrent fever occurred only in one of the children. The therapies included L-thyroxine, methotrexate, etanercept, insulin, glucocorticoids, and immunoglobulins. The treatments were ongoing at the time of publication, but the patients seemed to respond well [57].

Gln593Ter was found in a child with BD and lupus-like diseases. She had cervical lymphadenopathy and meningitis-like symptoms (fevers, headache, and neck pain). Between 8 and 11 years of age, she presented with malar rash, cervical lymphadenopathy, and proteinuria. Her condition improved after being treated with hydroxychloroquine and prednisone [58].

Thr602fs*95 was reported in a Chinese patient with Neuropsychiatric Systemic Lupus Erythematosus. The patient had different symptoms at the age of 36, including short-term memory loss, headache, cough, fever, and limb myalgia. Initially, she had polyarthritis, pancytopenia, and tonic–clonic epileptic seizures. She received different treatments, including intrathecal injections of dexamethasone and cyclophosphamide, which were effective [59].

A proband patient with Thr604Arg fs*93 had severe gastrointestinal ulcerations at the age of 15. Her family history was positive. Tofacitinib monotherapy was used for treatment, which the patient tolerated well, and improvement was seen in her condition [49].

A patient with Leu626Val fs*45 had autoinflammatory liver disease, which appeared at the age of 8. Tofacitinib monotherapy was successful, and hepatic transaminase levels were reduced in the patient [60].

His636fs*1 was associated with a case of adult-onset Still’s disease in a patient who had anorexia at the age of 16 and abdominal pain. Her C-reactive protein levels were high. She received treatment for urinary sepsis. Later, the patient developed additional symptoms, such as polyarthritis, fever, hyperferritinemia, and erythematous skin rash. Treatment with the anti-IL6 receptor biologic tocilizumab (RoActemra) seemed to be successful. The patient’s children developed inflammatory symptoms, such as malaise, fever, and abdominal pain. The patient’s brother also had fever, salmon-pink skin rashes, and vomiting. The affected family members were successfully treated with colchicine [61].

Glu730Serfs*83 was observed in a family where the affected patients developed intermittent fever and oral ulcers between the ages of 6 months and 10 years. One of them had a genital ulcer. Gastrointestinal involvement was observed in both patients (such as abdominal pain, IBD, and ulcers in the digestive tract). Both of them had short stature. The treatment included prednisone, infliximab, thalidomide, or azathioprine [34].

Gln737Serfs*79 was associated with a familial case of autoinflammatory disease. The proband had severe abdominal pain, febrile episodes, vomiting, and bloody stools, which occurred after 6 months of age. Other relatives also had autoinflammatory symptoms, such as recurrent fever, inflammatory bowel disease, stomatitis, and cervical lymphadenitis. Several family members, such as the mother, uncle, and grandmother of the proband, were diagnosed with Hashimoto’s thyroiditis. This study also suspected that A20 haploinsufficiency may be a risk factor for Hashimoto’s thyroiditis. They received levothyroxine and anti-thyroid autoantibody treatments [62].

Lys759Ser fs*56 was observed in a child with BD, who developed disease phenotypes starting from 5 months of age. The main symptoms were recurrent fever, hematochezia, steatorrhea, and elevated CRP, IL6, and IFN-alpha levels. Other family members, such as the great-grandmother and grandmother, also had BD or BD-like phenotypes [63].

### 2.3. Missense Mutations

Besides the splice site, frameshift, and nonsense variants, missense mutations have also been discovered in patients with A20 haploinsufficiency or other autoimmune diseases (Table 3, Figure 3).

The Asn102Ser mutation was found in a Chinese case of intestinal BD, and her first symptoms, such as abdominal pain, oral ulcers, and diarrhea, occurred at the age of 15. The patient also experienced recurrent colonic and anatomic ulcers and vasculitis in her extremities. Her CRP and erythrocyte levels were high. The patient had a positive family history of the disease since her father and brother had BD-related symptoms, such as oral ulcers, fever, and skin rash. Treatment with glucocorticoid and thalidomide reduced the abdominal symptoms [24].

A patient with Thr129Met was diagnosed with hemophagocytic lymphohistiocytosis (HLH) at the age of 4 months. His main symptoms were fever, irritability, pallor, pancytopenia, and hepatosplenomegaly. No other affected family members were found. He did not receive treatment with etanercept due to the high cost. Unfortunately, the patient’s condition deteriorated (cerebral hemorrhage; later, he was in a comatose state), and he passed away [64].

Glu192Lys was reported in a 4-year-old child who developed repeated tonsillitis, cervical lymphadenitis, and bronchitis before turning 1 year old. His serum CRP levels were high, and he was suspected to have an immunodeficiency disease. Treatment with prednisolone was initially successful for the fever. His final diagnosis was periodic fever, aphthous stomatitis, pharyngitis, and adenitis (PFAPA) syndrome. However, after 4 years of age, he experienced fever, temporary nonbacterial coxitis, and pharyngeal redness, and his treatment needed to be changed. Tonsillectomy surgery resulted in an improvement in his condition, as his fever episodes were reduced. The proband’s father carried the same variant and had BD-like symptoms, such as stomatitis, folliculitis, and hemorrhoids. The same study described two other TNFAIP3 variants (Ile310Thr and Gln709Arg), but their pathogenic nature remained unclear. These variants appeared in patients with inflammatory diseases. However, these variants did not segregate with the disease, as they were found in asymptomatic family members too [65].

The proband patient with Leu236Pro presented their first autoinflammatory symptoms at 15 months of age. Recurrent fever and ulcers occurred every 2 weeks. Other symptoms were abdominal pain, diarrhea, persistent folliculitis, and asymmetric joint arthralgia. Her sister had recurrent fever, persistent ulcers, abdominal pain, and diarrhea from the age of 5. Colchicine treatment was successful in these two siblings. Their father had episodic fever and ulcers starting from the age of 15. He also experienced further symptoms, including abdominal pain, diarrhea, colonic ulcers, myalgia, and arthralgia in the knees and ankles. He received a treatment consisting of colchicine, oral corticosteroids, and azathioprine, which resulted in improvement in his condition [66].

Cys243Tyr was reported in a Japanese familial case of BD, and the patients had oral and genital ulcers, and erythema nodosum-like lesions in their skin. All affected family members developed disease before 20 years of age. The proband patient and his mother produced large amounts of pro-inflammatory cytokines after stimulation. Some relatives reacted well to glucocorticoid therapy, but not to colchicine. Cell studies also confirmed the pathogenic nature of mutation [67].

Pro247Leu was observed in a BD patient with chronic inflammation. The proband developed recurrent oral ulcers with fever in his teenage years. He also had nephrotic syndrome, erythema nodosum-like lesions on his lower extremities, and pseudofolliculitis. Several relatives, such as his mother, grandmother, aunt, and cousin, developed BD or BD-like phenotypes. Low doses of prednisolone seemed to be useful against the oral ulcers [68].

Another missense mutation in TNFAIP3, Leu275Pro, was described by the same study in a 3-month-old child with HLH. He experienced prolonged fevers and hepatomegaly. Laboratory tests revealed various symptoms, such as pancytopenia, hyperferritinemia, hypoalbuminemia, hypertriglyceridemia, and hypofibrinogenemia. Treatment with different drugs (etoposide, dexamethasone, etanercept, and cyclosporine) was successful, as he was asymptomatic at the age of 4 years [65].

Thr474Ala was found in a Saudi child who had a unique phenotype of chronic immune thrombocytopenia and vitiligo in his early teenage years. He developed ecchymosis and bruises on his legs. Depigmentation occurred near both eyes, on his upper and lower extremities, and also in his hair. He did not present any typical BD or BD-like symptoms such as fever, ulcers, organomegaly, or lymphadenopathy. Intravenous immunoglobulin temporarily improved his symptoms, but other therapies seemed to be ineffective [69].

Met476Ile was found in a Chinese child with intermittent fever, skin rashes, and a high erythrocyte sedimentation rate in his early teenage years. The patient also presented joint pain during fever periods, liver dysfunction, diffuse lymphadenopathy, and recurrent tonsillitis and ulcers in the oral system and intestines. Besides the BD-like phenotypes, the patient also had persistent EBV viremia. The mother of the patient presented a milder phenotype, and she was also positive for the mutation. Antibiotics did not work, but an acyclovir and prednisolone treatment resulted in an improvement in the symptoms [70].

Thr602Ser was found in a family with BD-like autoinflammatory disease. The proband patient had recurrent fever, ulcers in the oral and perianal system, conjunctivitis, and inflammation of the bowel. The proband patient’s mother and brother also presented similar phenotypes. Prednisone and colchicine partially improved his condition, but he still experienced fevers [71].

Thr647Pro was found in a Pakistani Indian family. The proband was an 8-year-old child who had a neurosarcoid-like but unclassified granulomatous neuroinflammatory disorder at the age of 8 years. She presented cognitive decline, focal seizures and hemiparesis, ataxia, and uveitis. She also occasionally had ulcers in the mouth. Oral treatment with baricitinib, which is a JAK1/JAK2 inhibitor drug, was effective [72].

## 3. Possible Disease-Related Pathways and Functional Studies on TNFAIP3 Gene

As mentioned before, mutations in TNFAIP3 are associated with early-onset autosomal dominant autoinflammatory diseases [74]. Mutations in TNFAIP3 were confirmed to affect immune function through loss-of-function mechanisms. The A20 protein is an inhibitor of NFkappaB signaling, which is essential in controlling the expression of anti-apoptotic genes, in the production of inflammatory molecules, and in innate and adaptive immunity. NF-kappaB can be activated by different receptors including TNFRs (through RIPK1, receptor-interacting protein 1), Toll-like receptors (TLRs), or interleukin-1R (IL1R) receptors, leading to the activation of TNF receptor-associated factor 6 (TRAF6). Signal activation results in the activation of the IKK complex (IKK-alpha, IKK-beta, and IKK-gamma), and phosphorylation and degradation of IκB (NFkB inhibitor). This process results in the translocation of NFkappaB protein and increased production of pro-inflammatory cytokines [75,76,77].

A20 protein acts as a protective factor against the toxicity induced by TNF by removing the K63 and ubiquitin chains from RIPK1 and TRAF6, and inducing the K48 phosphorylation of the RIPK1 protein. A20 may also inhibit NFkappaB through non-enzymatic pathways. By interacting with the IKK complex, A20 may increase its stability and prevent its degradation and NFkappaB translocation. Furthermore, A20 can interact with A20-binding inhibitor of NF-kappaB (ABIN1), leading to IKK inhibition **(**Figure 4) [74,75,76,77,78].

Impairment in A20 function induces the NFkappaB pathways through deubiquitylation, higher IKK complex phosphorylation, and degradation of the NFkB inhibitor, leading to increased production of pro-inflammatory cytokines. Patients with TNFAIP3 mutations usually present elevated levels of pro-inflammatory molecules in their serum, such as IL1-beta, IFN-gamma, TNF-alpha, IL6, and IL18 during their acute phase. However, several cytokines remain higher than normal in asymptomatic periods too [79,80].

A20 protein was verified to be a regulator of apoptosis and necrosis since it was verified to have both pro-apoptotic and anti-apoptotic effects [81]. A20 was verified to protect against TNF-related cytotoxicity, but the exact mechanisms through which A20 induces or protects against cell death are not fully understood [81].

Priem et al. (2019) revealed that A20 knockout may result in RIPK1-dependent and RIPK1-independent apoptosis after stimulation with a low dose of TNF. A20 could protect cells from TNF-induced apoptosis through its ZNF7 domain, which could bind to M1-linked (linear) ubiquitin chains in respiratory complex I. Through this mechanism, A20 could protect against CYLD lysine-63 deubiquitinase (CYLD)-mediated apoptosis. In the case of M1-ubiquitin deficiency, A20 is also able to recruit complex I through its ZNF4 and ZNF7 domains. Without M1-linked ubiquitin, A20 ZNF4 and ZNF7 can bind some residual ubiquitin chains in complex I (such as K63-linked chains), and perform deubiquitination on a putative substrate, which has not been identified [81].

Lim et al. (2017) revealed that A20 might protect against cell death in the case of Helicobacter pylori infection by inhibiting NFkappaB activity and caspase-8–p62 complex formation. A20 inhibits the caspase-8 pathway through deubiquitylation by preventing the cullin3-mediated K63-ubiquitinylation of procaspase-8, preventing the activation of caspase-8 activity [82].

Onizawa et al. (2020) revealed that mouse models with abnormal A20 resulted in the K5-ubiquitinoylation of RIPK3, and the formation of RIPK1–RIPK3 complexes, leading to necroptosis. A20 may play a role in the prevention of this process through its C103 motif [83].

However, A20 could also have pro-apoptotic effects. Feoktistova et al. (2020) revealed that A20 can prevent cell death through regulating the TNF-induced cell death signaling pathways in keratinocytes. Elevated A20 expression may make keratinocytes more sensitive to TNF-induced apoptosis, increasing ripoptosome formation [84].

Zhou et al. (2016) performed NFκB luciferase assays on TNFAIP3 mutants (such as Phe224fs, Leu227Ter, Arg271Ter, Tyr306Ter, Pro268Leu fs19, and Thr604Argfs93). Cells derived from patients showed a higher degree of translocation of NFkappaB p65 into the nucleus after the degradation of NFkappaB inhibitors. This process could increase the level of NFkappaB-dependent pro-inflammatory cytokine production. Mutant TNFAIP3 may result in the cleavage of K63-ubiquitin and elevated TNF expression, and it may increase the expression of IKK-gamma, RIPK1, and TRAF6 proteins. Cells transfected with TNFAIP3 mutants were associated with an abnormal ubiquitylation/deubiquitylation patterns. The abnormal deubiquitylation of A20 protein could result in high NFkappaB signaling in the mutant cells [48].

Schwarz et al. (2021) studied three mutations in the TNFAIP3 gene using luciferase assay. Phe204fs4, Thr604Argfs93, and Leu626Valfs*45 were suggested to induce type I interferons (IFNs), leading to increased expression of IFN-stimulated genes (ISGs). The expression of ISGs in A20 haploinsufficiency could depend on NFkappaB, but they may be induced by alternative mechanisms too. Janus kinase (JAK) signaling inhibitors, which could target IFN signaling, could be used in the treatment of A20 haploinsufficiency [60].

Rajamäki et al. (2018) revealed that a nonsense mutation in TNFAIP3 (Lys91Ter) could result in crosstalk between NFkappaB signaling proteins, DNA repair proteins, and Nod-like receptor (NLR) family pyrin domain-containing 3 (NLRP3) inflammasome proteins. This process could lead to the maturation of caspase-1-mediated proteolysis and the production of IL1-beta and IL18 pro-inflammatory proteins. Inhibiting caspase-8 resulted in reduced production of these cytokines. This study suggested that the elevated secretion of IL1-beta and IL18 could be associated with a putative caspase-8-dependent mechanism. TNFAIP3 deficiency may result in elevated caspase-8 activity. Also, the Lys91Ter mutation may result in the loss of interaction between baculoviral IAP repeat-containing protein 2 (BIRC2) and the A20 protein, which could lead to increased caspase-8 activity and cell death [20].

Normally, the A20 protein inhibits signal transducer and activator of transcription 1 (STAT1) expression, which could regulate the expression of IFN-gamma-related genes and chemokines. The TNFAIP3 mutation His577Alafs*95 was suggested to increase the level of IFN-gamma-related chemokines, such as CXCL9 and CXCL10. The study suggested that TNFAIP3 mutations may be associated with an elevated degree of STAT1 and STAT3 phosphorylation [57].

## 4. TNFAIP3 in Atypical Inflammatory Diseases

As mentioned before, patients with A20 haploinsufficiency can present diverse phenotypes. Besides the classical A20 haploinsufficiency or BD phenotypes, additional atypical phenotypes have been reported in TNFAIP2 mutant patients, including diabetes, psoriasis, Crohn’s disease, IBD, systematic lupus erythematosus (SLE), lupus nephritis, and neurological dysfunction (Figure 5).

TNFAIP3 mutations, such as Arg87Ter, Val489Aal fs7, Gln490Ter, are His577Alafs95, were related to BD-like symptoms and type 1 (T1D) diabetes, suggesting that A20 haploinsufficiency may be related to the onset of diabetes. It was suggested that the expression of TNFAIP3 was up-regulated in the beta cells of the pancreatic islets. Animal experiments revealed that in the case of pancreatic islet transplantation, A20 may protect against graft rejection. Also, normal A20 functions may be essential for beta-cell survival and appropriate functioning. A deficiency of A20 was associated with abnormal glucose metabolism and insulin secretion, and low expression of regulatory genes for appropriate beta-cell functioning. Elevated NFkappaB levels may result in autoimmunity in pancreatic beta cells [56,85].

TNFAIP3 was also associated with psoriasis, a chronic inflammatory disease. In the blood of patients and mouse models for psoriasis, TNFAIP3 mRNA expression was reduced compared to unaffected individuals or control mice, respectively. In mouse and rat models, Th1 and Th17 cells and pro-inflammatory serum cytokines, including IL17a, IL23, IFN-gamma, and TNF-alpha, were increased. These cytokines induce the proliferation of keratinocytes and immune cell infiltration, leading to the development of psoriatic plaques. Moreover, phosphorylated p38 levels were higher in psoriasis mouse and rat models compared to control animals. Treating the mice and rats with TNF-alpha antagonists resulted in reduced pro-inflammatory cytokine levels and increased TNFAIP3 expression. These findings reveal that TNFAIP3 dysfunction and downregulation may impact psoriasis through impaired Th1 and TH17 cells and p38 activation [86].

TNFAIP3 may also impact the digestive system. Some patients with Crohn’s disease have reduced expression of TNFAIP3. TNFAIP3 was suggested to play a role in maintaining intestinal barrier function. TNFAIP3 knockout in mice may be associated with higher permeability of the intestinal barrier and a higher tendency toward TLR-mediated spontaneous inflammation. TNFAIP3 dysfunction was associated with a dysregulated myosin light-chain kinase (MLCK) pathway through TNF signaling. TNFAIP3 may not control MLCK activity directly, but it may impact the TNF/MLCK/tight junction pathway through an unknown mechanism. It may be possible that TNFAIP3 could play a role in the MLCK pathway in the intestinal barrier through ubiquitination. TNFAIP3 was found to regulate the localization of occludin (a tight junction protein) through deubiquitinating mechanisms. TNFAIP3 dysfunction resulted in the loss of occludin protein in the apical border of the intestinal epithelium [87]. TNFAIP3 may also impact inflammatory bowel diseases by interacting with autophagy-related 16-like 1 (ATG16L1) protein. This interaction may be possible through the OTU domain of TNFAIP3 protein and the WD40 domain of ATG16L1. Impairment of this interaction may impair the homeostasis of the intestinal system through different mechanisms, such as autophagy through sequestosome-1 (SQSTM1) degradation, and NFkappaB activation through TNFs and TNFAIP3 degradation [88]. A20 haploinsufficiency can also result in an imbalance in the gut microbiota. It may be possible that the altered interaction between the microbiome and the immune system may play a role in disease pathogenesis [89].

TNFAIP3 mutations were observed in lupus and lupus-like phenotypes, such as systematic lupus erythematosus (SLE) and lupus nephritis. An SLE patient with a TNFAIP3 mutation (Thr602fs95) had elevated levels of NFkappaB and mitogen-activated protein kinase pathway components. In patient-derived T-cells, the phosphorylation of p38 and JNK was increased, leading to the elevated expression of pro-inflammatory cytokines. The mutation resulted in decreased K63-ubiquitin levels, leading to the disruption of the interaction between TNF and TRAF6 and A20 [59]. TNFAIP3 mutations, such as Asp212Glyfs38 (c.634+2T>C) [30], exon 7–8 deletion, or p.Ala434Ter (c.1300_1301delinsTA) [69], were associated with lupus nephritis [90]. Further kidney dysfunctions were identified in patients with TNFAIP3 mutations, such as Glu187Ter and Phe224Serfs*4, whose presented with proliferative glomerulonephritis and membranous lupus nephritis, respectively [8,44]. These patients showed elevated NF-kappaB signaling and IFN-I activation, and they had elevated levels of pro-inflammatory cytokines in the blood [30,91]. The exact mechanism through which A20 haploinsufficiency impacts lupus nephritis or other kidney dysfunction disorders remains unclear. It is possible that A20 protein interacts with other genetic factors involved in lupus nephritis [30,90]

In rare cases, TNFAIP3 mutations have been associated with neurological dysfunction. For example, TNFAIP3 Thr602fs*95 was linked to neurological dysfunction in addition to SLE. The mutation was suggested to destroy the tight junctions in the blood–brain barrier (BBB), leading to the infiltration of pro-inflammatory cytokines into the central nervous system. Furthermore, the mutation could result in elevated microglia activation and a cytokine imbalance in the brain [59]. TNFAIP3 was also suggested to play a role in multiple sclerosis (MS), as TNFAIP3 was downregulated in the blood of MS patients. However, post-mortem brain analysis of A20 expression in human brains revealed that A20 expression was high in immune cells, such as infiltrating macrophages, activated astrocytes, and microglia. A20 expression was high in MS plaques, and active and chronically active white matter lesions in MS patients. These studies suggest that TNFAIP3 may impact MS, but the underlying mechanisms remain unclear [92].

## 5. Therapeutic Strategies for A20 Haploinsufficiency

Immune dysfunctions associated with A20 haploinsufficiency can present as diverse clinical phenotypes, as multiple organs may be affected by the disorder. There is no standard treatment protocol available for the disease itself, and the current strategies usually focus on the inflammation-related symptoms. The most frequently used treatments include corticosteroids or other anti-inflammatory drugs, such as TNF-alpha inhibitors, colchicine, or methotrexate. Additional immunosuppressive drugs, such as azathioprine, and biological agents such as anakinra, rituximab, and tocilizumab, are also commonly used [10,93].

Corticosteroids have been verified as effective treatments for inflammatory diseases, as they suppress inflammation by reducing the degree of histone acetylation and preventing the expression of inflammatory genes [94]. The majority of patients with A20 haploinsufficiency respond well to corticosteroids, but they must be administered in high doses, resulting in side effects such as diabetes, arterial hypertension, or growth issues [8].

Colchicine has been used to treat multiple autoimmune diseases. This drug acts through different pathways, such as preventing microtubule formation; inhibiting chemotaxis, leading to the activation anti-inflammatory pathways; or preventing inflammasome formation. In A20 haploinsufficiency-related diseases, such as BD, colchicine is used to treat ulcers or articular disease phenotypes. Colchicine may act by inhibiting chemotaxis and reducing the production of pro-inflammatory cytokines. However, a large dose of colchicine is associated with an increased risk of vascular dysfunction, such as thrombosis [8,95].

TNF-alpha blockers, such as infliximab, adalimumab, and etanercept, are also common therapies for BD and other inflammatory diseases. These drugs usually focus on reducing TNF-alpha production by immune cells, such as macrophages, CD4+ and CD8+ T-cells, and NK cells [96,97]. However, this treatment can result in side effects, including a psoriasis-like syndrome. It is possible that combining anti-TNF therapy with the induction of IL17A production would be an effective treatment for A20 haploinsufficiency [98].

Methotrexate (MTX) treatment has been successfully used to treat the neuropsychiatric symptoms or ocular dysfunction of BD patients [99,100]. Low-dose MTX treatment reduced the IL6 levels in the CSF of patients with BD; however, after discontinuation of treatment, a mild reduction in their condition was observed. The study suggested that low-dose MTX therapy should be given for a longer period [13].

Thalidomide is also a widely used drug in BD treatment, and in several cases, it has been used successfully. The effect of thalidomide may be associated with inhibiting the migration of neutrophil granulocytes and vessel damage, and reducing the TNF levels in patients. The drug was successfully used against oral and genital ulcers. Also, thalidomide was effectively used in patients who developed BD at young ages [101,102].

A20 was verified to suppress NLRP3 inflammasome formation. In the case of A20 deficiency, IL-beta K-133 ubiquitinoylation was increased in addition to NLRP3 activation [103]. It was suggested that anti-IL-1 drugs, such as canakinumab or anakinra, may also be promising therapeutics against BD. These drugs may be generally safe options for first-choice therapy in BD patients [104]

Baricitinib is a JAK1/2 inhibitor, which has been effectively used in autoinflammatory diseases, since inflammatory markers, especially interferons, were improved in the patients [105]. Baricitinib was also successfully used to treat A20 haploinsufficiency since it improved the type I interferon responses and reduced the expression of IFN-stimulated genes in the patients [106].

## 6. Discussion

A20 haploinsufficiency is an immune-related disease caused by mutations in the TNFAIP3 gene. The disease can affect multiple organs, including the skin, intestines, liver, and, in rare cases, the brain. Several mutations have been described in the TNFAIP3 gene, including splice site, STOP codon, and frameshift variants. Missense mutations may also be associated with a loss or reduction of A20 function [68]. Mutant TNFAIP3 results in reduced A20 function, leading to the inhibition of IkappaB kinase and elevated phosphorylation of NFkappaB. This leads to the elevated production of pro-inflammatory molecules (such as IL1-beta, IL2, IL6, TNF-alpha, and IL8) and the dysfunction of multiple organ [68].

Several phenotypes have been identified in A20 haploinsufficiency patients. The majority of patients were diagnosed with BD or juvenile idiopathic arthritis at young ages, but late-onset cases have also been identified. However, patients with TNFAIP3 mutations presented additional autoinflammatory disease phenotypes, including systemic lupus erythematosus, psoriatic arthritis, autoimmune hepatitis, nephrotic syndrome, and Hashimoto’s thyroiditis. The location of the mutations may influence the clinical phenotype. Mutations in the OTU or ZNF domains can both impact the autoinflammatory phenotypes. The loss of the OTU and ZNF domains, or the loss of the ZNF domains alone, are more likely to be associated with musculoskeletal dysfunction compared to the loss of the OTU domain alone. Patients with OTU mutations are more likely to be diagnosed with BD compared to those with mutations in the ZNF domains. It has been suggested that mutations in the ZNF domains may be associated with an earlier age of onset compared to the mutations in the OTU domain [72,107].

Other factors may also explain the diverse symptoms of A20 haploinsufficiency cases. The gender of patients could possibly affect the clinical manifestation of the disease. Female patients were suggested to have a higher risk of developing genital ulcers, while male patients are more likely to present with ulcers in the gastrointestinal system or perineal inflammation [93]. Furthermore, environmental factors (diet or chemical exposure) or putative genetic risk modifiers were also suggested to impact the clinical impact of TNFAIP3 mutations [108].

In terms of treatment, it is important to note that the effectiveness of therapies can vary among individuals due to the genetic and environmental factors that influence the severity of HA20. Future research is needed to establish methods for analyzing the pathophysiology of HA20 and determining the optimal treatment strategies for each patient [10,93,108].

In conclusion, patients with suspected autoinflammatory symptoms should be tested for TNFAIP3 mutations. The pathogenic nature of the missense mutations should not be overlooked; however, functional studies may be needed to verify their role in disease progression. Further studies are also necessary to explore the mechanisms of mutations located in different A20 domains to optimize the therapies for inflammatory diseases [68,72].

## Figures and Tables

**Figure 1 ijms-25-08275-f001:**
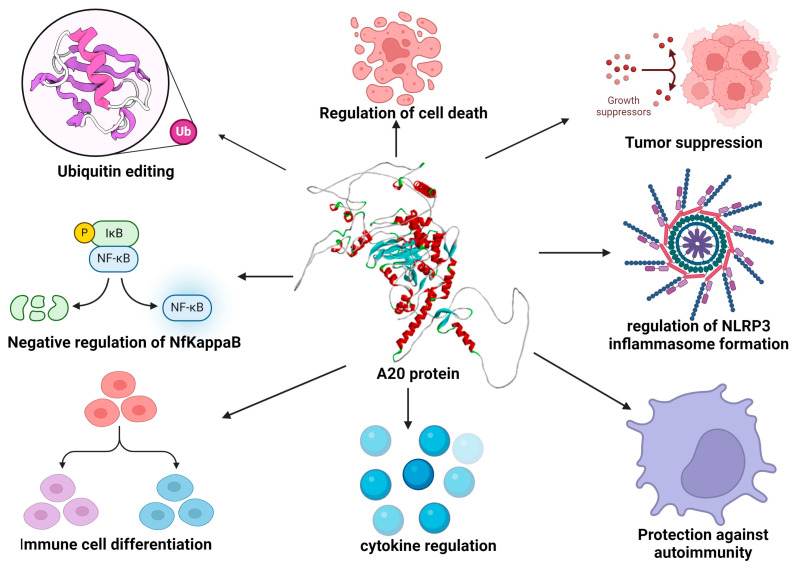
Functions of TNFAIP3 (A20) protein [1,2,3,4,5,6]. The 3D structure of the A20 protein was generated using the Alphafold Colab v1.5.5 tool [7] (https://colab.research.google.com/github/sokrypton/ColabFold/blob/main/AlphaFold2.ipynb, accessed 10 June 2024). Figure was made using the BioRender tool.

**Figure 2 ijms-25-08275-f002:**
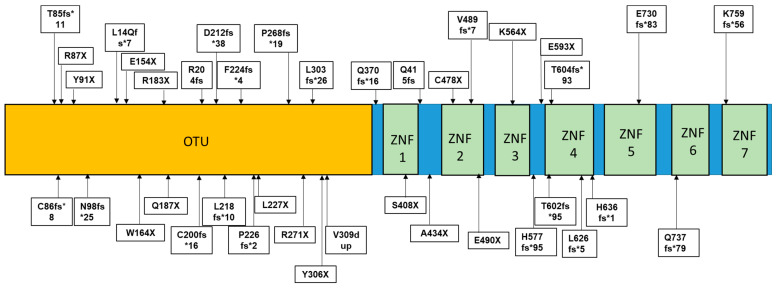
Frameshift and STOP codon mutations in TNFAIP3 gene.

**Figure 3 ijms-25-08275-f003:**
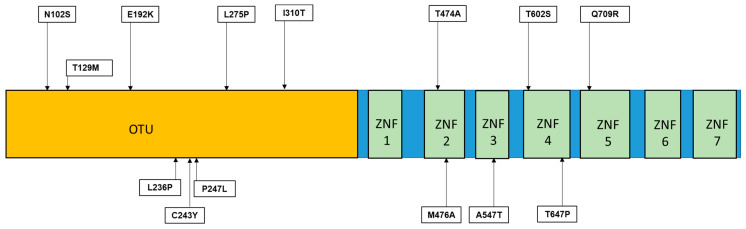
Missense mutations in TNFAIP3 gene.

**Figure 4 ijms-25-08275-f004:**
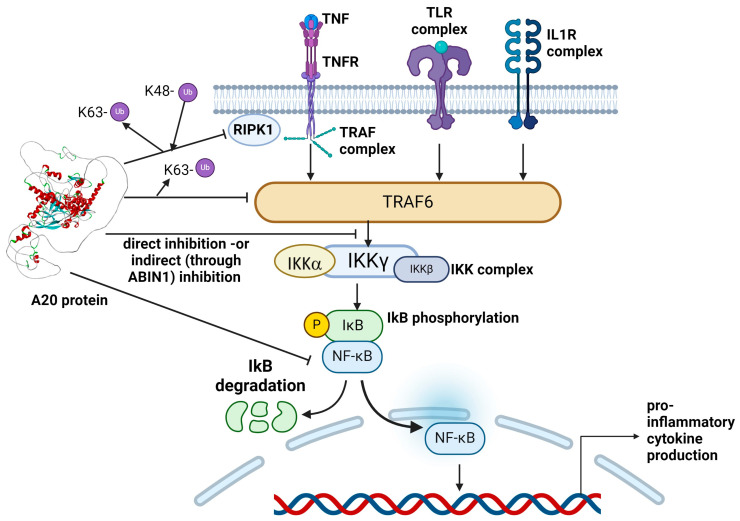
Roles of A20 protein in NfkappaB signaling [61,62,63]. The 3D structure of the A20 protein was generated using Alphafold Colab v1.5.5 tool (https://colab.research.google.com/github/sokrypton/ColabFold/blob/main/AlphaFold2.ipynb, accessed on 10 June 2024) [7]. Figure was made using BioRender tool.

**Figure 5 ijms-25-08275-f005:**
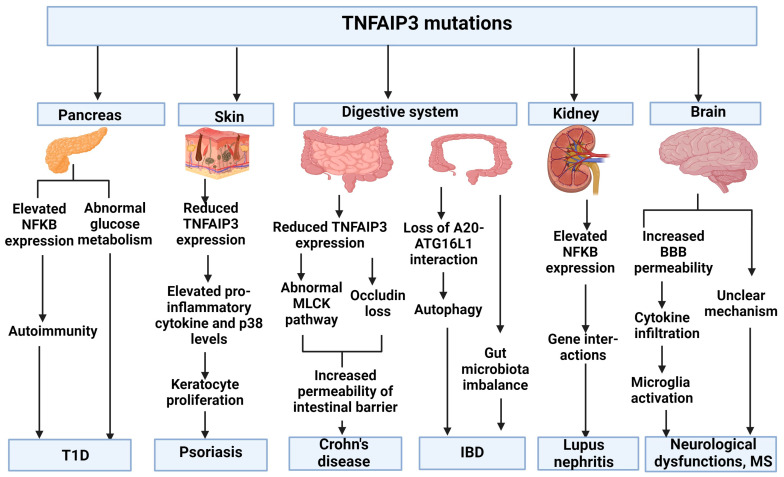
Possible pathways affected by TNFAIP3 mutations associated with atypical phenotypes. Figure was made using BioRender Tool.

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
