# Peer review of "Genetic Mutations Associated With TNFAIP3 (A20) Haploinsufficiency and Their Impact on Inflammatory Diseases"

_ijms, 2024, doi:10.3390/ijms25158275_

Round 1

Reviewer 1 Report

Comments and Suggestions for Authors

In this manuscript, the authors present a review about the crucial role of TNF-α-induced protein 3 (TNFAIP3) in immune regulation, apoptosis, and initiating immune responses.

Mutations in TNFAIP3 are linked to immune-related diseases such as Behçet’s disease, juvenile idiopathic arthritis, autoimmune thyroiditis, autoimmune hepatitis, and rheumatoid arthritis, leading to symptoms like recurrent fever, ulcers, rashes, musculoskeletal, gastrointestinal, cardiovascular, and respiratory issues.

Most mutations are nonsense or frameshift, causing immune dysfunctions, but missense mutations also contribute. These mutations disrupt pathways like NF-κB signaling and ubiquitination. No definitive treatment exists for A20 haploinsufficiency, but therapies can alleviate symptoms. This review covers TNFAIP3 mutations, clinical progression, disease mechanisms, and available treatments. 

1) The manuscript is well written, but it is hard to read. In my opinion, it should be divided into thematic paragraphs to make it easier for the reader to understand.  

2) The overall writing has some formatting issues, like wording and spacing. I suggest the authors check the grammar and avoid any typos.

3) In the legend of Figure 1 and Figure 4 authors have to indicate the reference of Alphafold Colab v1.5.5 tool (Mirdita M, Schütze K, Moriwaki Y, Heo L, Ovchinnikov S, Steinegger M. ColabFold: Making protein folding accessible to all. Nature Methods, 2022) and not simply the link to download the tool.

4) In Lines 607, 651, 707 please remove PMID number

5) I suggest that the authors create figures of high resolution.

I find the work highly engaging, and it offers a valuable overview of the current state of the field.

Author Response

In this manuscript, the authors present a review about the crucial role of TNF-α-induced protein 3 (TNFAIP3) in immune regulation, apoptosis, and initiating immune responses.

Mutations in TNFAIP3 are linked to immune-related diseases such as Behçet’s disease, juvenile idiopathic arthritis, autoimmune thyroiditis, autoimmune hepatitis, and rheumatoid arthritis, leading to symptoms like recurrent fever, ulcers, rashes, musculoskeletal, gastrointestinal, cardiovascular, and respiratory issues.

Most mutations are nonsense or frameshift, causing immune dysfunctions, but missense mutations also contribute. These mutations disrupt pathways like NF-κB signaling and ubiquitination. No definitive treatment exists for A20 haploinsufficiency, but therapies can alleviate symptoms. This review covers TNFAIP3 mutations, clinical progression, disease mechanisms, and available treatments. 

Thank you very much for the encouraging comments. We revised the manuscript, according to your suggestions.

  • The manuscript is well written, but it is hard to read. In my opinion, it should be divided into thematic paragraphs to make it easier for the reader to understand. 

Thank you, we re-divided the paragraphs in the manuscript for better understanding. 

  • The overall writing has some formatting issues, like wording and spacing. I suggest the authors check the grammar and avoid any typos.

Thank you, we tried to fix the typos and grammar.

  • In the legend of Figure 1 and Figure 4 authors have to indicate the reference of Alphafold Colab v1.5.5 tool (Mirdita M, Schütze K, Moriwaki Y, Heo L, Ovchinnikov S, Steinegger M. ColabFold: Making protein folding accessible to all. Nature Methods, 2022) and not simply the link to download the tool.

Thank you, we added the requested reference to the manuscript.

  • In Lines 607, 651, 707 please remove PMID number

Thank you, we removed PMID numbers

  • I suggest that the authors create figures of high resolution.

Thank you, The figures were recreated and resent in better resolution.

Reviewer 2 Report

Comments and Suggestions for Authors

A20 is a protein encoded by the TNFAIP3 gene, A20 regulates many signaling pathways critical for inflammation, proliferation, and cellular homeostasis.

In this review, the authors provide an overview of case reports of disease-related mutations found in the coding region of the TNFAIP3 gene, detailing their symptoms, treatment, and disease progression. Additionally, they discuss potential A20-regulated pathways that could be associated with disease.

Overall, the manuscript is quite detailed and well-written. However, I have some minor comments.

1)      Around line 88, this could be a good moment to describe haploinsufficiency as you did at the beginning of the discussion.

2)      In lines 90-93, there is a duplicate sentence.

3)      The section Possible Disease Related Pathways and Functional Studies on TNFAIP3 Gene could be enriched by discussing the role of A20 in cellular apoptosis and necroptosis.

Priem D, et al. A20 protects cells from TNF-induced apoptosis

through linear ubiquitin-dependent and -independent mechanisms.

Cell Death Dis. 2019;10:692.

Onizawa M, et al. The ubiquitin-modifying enzyme A20 restricts

ubiquitination of the kinase RIPK3 and protects cells from necroptosis.

Nat Immunol. 2015;16:618–27.

Lim MCC, et al. Pathogen-induced ubiquitin-editing enzyme A20

bifunctionally shuts off NF-κB and caspase-8-dependent apoptotic cell death. Cell Death Differ. 2017;24:1621–31.

4)      This article could be cited in the introduction.

Martens A, et al. Two distinct ubiquitin-binding motifs in A20 mediate its anti-inflammatory and cell-protective activities. Nat Immunol. 2020;21:381–7 

5)      Thalidomide can also be added to line 633. Also, it could be helpful to remark on why mechanistically anti-IL-1 therapies are effective in some patients (A20 also suppresses the NLRP3 inflammasome) and why Jak inhibitors could also be used.

Author Response

A20 is a protein encoded by the TNFAIP3 gene, A20 regulates many signaling pathways critical for inflammation, proliferation, and cellular homeostasis.

In this review, the authors provide an overview of case reports of disease-related mutations found in the coding region of the TNFAIP3 gene, detailing their symptoms, treatment, and disease progression. Additionally, they discuss potential A20-regulated pathways that could be associated with disease.

Overall, the manuscript is quite detailed and well-written. However, I have some minor comments.

 Thank you for the positive and encouraging comments. We tried to fix the manuscript, according to your suggestions.

1)      Around line 88, this could be a good moment to describe haploinsufficiency as you did at the beginning of the discussion.

 Thank you, we agree your suggestion, and we fixed this issue. We moved the paragraphs on haploinsufficiency in the introduction.

2)      In lines 90-93, there is a duplicate sentence.

 Thank you very much, this issue has been fixed.

“Up to date, 12 cases of large deletion relation-related A20 haploinsufficiency have been reported.”

3)      The section Possible Disease Related Pathways and Functional Studies on TNFAIP3 Gene could be enriched by discussing the role of A20 in cellular apoptosis and necroptosis.

Priem D, et al. A20 protects cells from TNF-induced apoptosis

through linear ubiquitin-dependent and -independent mechanisms.

Cell Death Dis. 2019;10:692.

Onizawa M, et al. The ubiquitin-modifying enzyme A20 restricts

ubiquitination of the kinase RIPK3 and protects cells from necroptosis.

Nat Immunol. 2015;16:618–27.

Lim MCC, et al. Pathogen-induced ubiquitin-editing enzyme A20

bifunctionally shuts off NF-κB and caspase-8-dependent apoptotic cell death. Cell Death Differ. 2017;24:1621–31.

 Thank you very much for your suggestions, they are fascinating studies. I added a chapter, which included these (and other) studies.

“ A20 protein was verified to be a regulator of apoptosis and necrosis since it was ver-ified to have both pro-apoptotic and anti-apoptotic effects [PMID: 31534131, PMID: 32028675]. A20 was verified to protect against TNF-related cytotoxicity, but the exact mechanisms, of how A20 could induce or protect against cell death is not fully understood yet. [PMID: 32241683, PMID: 31534131, PMID: 31974345].

 Priem et al. (2019) revealed that A20 knockout may result in RIPK1-dependent and RIPK1-independent apoptosis after stimulating with a low dose of TNF. A20 could protect cells from TNF-induced apoptosis by its Znf7 domain, which could bind to M1-linked (linear) ubiquitin chains in respiratory complex-I. By this step, A20 could protect against CYLD lysine 63 deubiquitinase (CYLD)-mediated apoptosis. In case of M1-ubiquitin deficiency, A20 is also able to recruit the Complex-I through its Znf4 and Znf7 domains. Without M1-linked ubiquitin, A20 Znf4 and Znf7 can bind some residual ubiquitin changes in Complex-I (such as K63-linked chains), and perform deubiquitination on a putative substrate, which was not identified yet [PMID: 31534131, PMID: 31974345].

Lim et al. (2017) revealed that A20 might protect against cell death in the case of Helicobacter pylori infection by inhibiting the NFkappaB activity and caspase-8-p62 com-plex formation. A20 inhibits the Caspase-8 pathway through deubiquitylation by preventing the cullin3-mediated K63-ubiquitinylation of procaspase-8, leading to the prevention of caspase-8 activity [PMID: 28574503].

Onizawa et al. (2020) revealed that mouse models with abnormal A20 resulted in the K5- ubiquitinoylation of RIPK3, and the formation of RIPK1-RIPK3 complexes, leading to necroptosis. A20 may play a role in the prevention of this process through its C103 motif.  [PMID: 25939025].

However, A20 could also have pro-apoptotic effects. Feoktistova et al. (2020) revealed that A20 cell death through TNF-induced cell death signaling pathways in the keratinocytes. Elevated A20 expression may make the keratinocytes more sensitive to TNF-induced apoptosis, increasing the ripoptosome formation [PMID: 32028675].”

4)      This article could be cited in the introduction.

 Martens A, et al. Two distinct ubiquitin-binding motifs in A20 mediate its anti-inflammatory and cell-protective activities. Nat Immunol. 2020;21:381–7

Thank you, we cited this article.

5)      Thalidomide can also be added to line 633. Also, it could be helpful to remark on why mechanistically anti-IL-1 therapies are effective in some patients (A20 also suppresses the NLRP3 inflammasome) and why Jak inhibitors could also be used.

Thank you for the suggestion. We added your suggestion to Chapter 5.

“Thalidomide has also been a widely used drug in BD treatment, and in several cases, it has been used successfully. The effect of thalidomide may be associated with inhibiting the migration of neutrophil granulocytes, vessel damage, and reduction of TNF levels in patients. The drug was successfully used against oral or genital ulcers. Also, thalidomide was effectively used in patients, who developed BD at young ages [PMID: 11908493, PMID: 36367251].

A20 was verified to suppress the NLRP3 inflammasome formation. In the case of A20 deficiency, the IL-beta K-133 ubiquitinoylation was increased beside the NLRP3 activation [PMID: 25607459]. It was suggested that anti-IL-1 drugs, such as canakinumab or anakinra may also be promising therapy against BD. These drugs may be generally safe options for first-choice therapy in BD patients [PMID: 26156661]

Baricitinib is a JAK1/2 inhibitor, which has been verified to be effectively used in autoinflammatory diseases, since the inflammatory markers, especially interferons were improved in the patients [PMID: 29649002]. Baricitinib was also successfully used in A20 haploinsufficiency since it improved the Type-I interferon responses and reduced expression of the IFN-stimulated genes among the patients [PMID: 31767699]”

Round 2

Reviewer 1 Report

Comments and Suggestions for Authors

The authors have improved the manuscript based on my suggestions. The text still remains heavy to read, but I understand that the descriptive content, especially regarding the individual mutations, is not easy to simplify.